# Academic Achievements, Satisfaction with Studies and Risky Behaviours among First-Year Students of Kaunas (Lithuania) Universities, 2000–2017

**DOI:** 10.3390/ijerph19137616

**Published:** 2022-06-22

**Authors:** Janina Petkeviciene, Vilma Kriaucioniene, Asta Raskiliene

**Affiliations:** 1Health Research Institute, Faculty of Public Health, Lithuanian University of Health Sciences, LT47181 Kaunas, Lithuania; vilma.kriaucioniene@lsmuni.lt (V.K.); asta.raskiliene@lsmuni.lt (A.R.); 2Department of Preventive Medicine, Faculty of Public Health, Lithuanian University of Health Sciences, LT47181 Kaunas, Lithuania

**Keywords:** alcohol consumption, CAGE test, smoking, drug use, academic achievements, trends, associations, students

## Abstract

Risky behaviours are prevalent among university students and may affect academic achievements. This study aimed to evaluate the associations between academic achievements, satisfaction with studies and risky behaviours among first-year students of Kaunas (Lithuania) universities. Three cross-sectional surveys were conducted in 2000, 2010, and 2017. The self-administered questionnaires were filled in during lectures and assessed frequency and amount of alcohol consumption, drinking problems (CAGE test), smoking and drug use frequency, and academic achievements. The associations between risky behaviours and academic achievements were analysed using multivariable logistic regression analysis. Altogether 3325 students (1341 men and 1984 women) aged 20.0 (1.5) years participated in the survey. The proportion of students who consumed alcohol at least once a week and drank 11 or more standard alcoholic units (SAU) a week decreased. Every fifth student reported a drinking problem. Daily smoking prevalence declined, and drug use increased among male students. Multivariable logistic regression analysis revealed that frequent alcohol consumption and problem drinking were associated with low importance of good grades. Students who rated their academic performance below average and were dissatisfied with studies were more likely to consume ≥11 SAU a week. Daily smoking was more common among students who reported low importance of good grades and academic performance below average. A higher prevalence of drug use was found only in male students who declared low importance of good grades. Health promoting interventions at the individual and student community level are required to reduce the prevalence of substance use and improve the academic achievements of students.

## 1. Introduction

Academic achievements are an important goal for students because the competitive job market requires advanced knowledge and skills. Students with good academic performance are more likely to become successful in their work careers and are more confident about their future [1]. This affects health and quality of life. Research evidence suggests that academic achievements of students are associated with various factors, including socioeconomic status, quality of an educational institution, learning strategies, learning motivation, cognitive abilities, and health behaviours, especially sleep and substance use [2,3,4]. Determining factors associated with academic achievements is important to universities and students.

First-year students represent a highly vulnerable population. The transition from high school to university is accompanied by life changes. Many students leave the parental home and face a variety of challenges such as new relationships with peers and lecturers, the need for independence and social approval, academic requirements, and managing finances [5,6]. These challenges can increase the stress that some students find difficult to overcome [7]. It is well-known that a healthy lifestyle promotes well-being and reduces health problems, including mental disorders [8]. Unfortunately, the stressors could lead to risky behaviours, such as smoking, alcohol and drug use, which prevalence tends to increase among first-year students [9,10,11].

Early adulthood is a sensitive period for substance use because significant physical and social changes occur during this life period. Alcohol, cannabis, opioids and nicotine are associated with neurocognitive alterations in the brain that contribute to behavioural consequences ranging from increased addiction risk to cognitive dysfunctions [12,13]. Thus, substance use may affect the intellectual development of students and decrease attentiveness and memory.

Previous studies indicated that hazardous alcohol consumption, smoking, and marijuana use were linked to low academic performance [4,14,15]. Students with risky behaviours had lower academic motivation, received lower grades, and failed more subjects than students with a healthy lifestyle [16,17].

The prevalence of risky behaviours is high in Lithuania. Despite the fact that alcohol consumption has decreased in recent years, Lithuania remains in the upper quintile in levels of consumption per capita [18]. Based on the European Health Interview Survey, daily smoking prevalence decreased from 42.1% in 2005 to 29.9% in 2019 among Lithuanian men and remained stable among women (9.8% in 2005 and 9.7% in 2019) [19]. Smoking and alcohol consumption are highly prevalent also in Lithuanian higher education institutions [20]. A recent study carried out in universities and colleges of Lithuania demonstrated that during the last month, 33% of respondents smoked cigarettes, 59% drank beer, 35% consumed strong alcoholic beverages, and 11% tried marijuana/hashish at least once [21]. However, data on the effect of substance use on the academic achievements of students are lacking in Lithuania. Also, no study analysed time trends in students’ risky behaviours using the same methodology.

This study aimed to evaluate the associations between academic achievements, satisfaction with studies and risky behaviours among first-year students of Kaunas (Lithuania) universities using data from the surveys carried out in 2000, 2010 and 2017.

## 2. Materials and Methods

### 2.1. Study Design and Sample

The data were obtained from three cross-sectional surveys carried out among the first-year students of five Kaunas (Lithuania) universities in 2000, 2010 and 2017. The participating universities were: Kaunas University of Technology, Vytautas Magnus University, Lithuanian University of Health Sciences, Lithuanian Sports University, and Agriculture Academy. First-year students from the selected faculties were invited to complete a self-administered questionnaire during the lectures. The faculties were selected to ensure that the major disciplines, such as health sciences, engineering, social sciences, humanities, and business, were represented in the study sample. Researchers visited the lectures to provide information about the study and invite students to complete the questionnaire. Students filled in the questionnaires at the end of the lecture. Participation was anonymous and voluntary, and response rates were 97%, 82%, and 96% in 2000, 2010, and 2017, respectively. In total, data of 3325 students (1341 men and 1984 women) were included in the analysis. 

Permission to conduct the study was obtained from the administration of the universities. The study protocol was approved by the Bioethics Centre at the Lithuanian University of Health Sciences (BC-VSF(M)-186; BC-VSF(M)-185; BEC-VSF(M)-84). Written informed consent was signed by all participating students.

### 2.2. Questionnaire

The same questionnaire was used in all studies. The questionnaire included a range of questions on risky behaviours. Students were asked about the frequency of alcohol drinking in the last three months. Possible answers were: ‘several times per day’, ‘every day’, ‘several times per week’, ‘once a week’, ‘several times or once a month’, ’less often or never’). For data analysis, a dichotomous variable was created: ‘drinking alcohol at least once a week’ and ‘drinking less often’. The amount of alcohol consumed was assessed by asking the question: ‘How many alcoholic drinks do you consume in a normal week (glasses of beer (200 mL), glasses of wine or sparkling wine (100 mL), glasses of strong alcoholic beverage such as vodka, whiskey, brandy, etc. (40 mL)?’ The answers to this question were used for the calculation of standard alcoholic units (SAUs). In Lithuania, one SAU contains 10 g of pure alcohol. Calculations were performed for each type of alcoholic beverage separately using the formula: SAUs = (consumed volume (mL) × strength in percentage/100 × 0.8)/10.

In this formula, strength is the amount of pure alcohol as a percentage of the total volume: for beer—5%, for wine—12%, for strong alcoholic beverage—40%. A factor of 0.8 was used to convert the pure alcohol content from ml to grams. The sum of the SAUs obtained from each type of alcoholic beverage was calculated as the total number of SAUs consumed per week. This variable was dichotomized using 11 SAUs as the cut-off (≥11 SAUs and <11 SAUs). This cut-off was chosen following the guidelines of the United Kingdom (UK) that provide advice for men and women not to drink more than 14 units of alcohol a week regularly [22]. In the UK, one SAU contains 8 g of pure alcohol. Converted to 10 g of pure alcohol used for one SAU in Lithuania, 14 UK units became equal to 11 Lithuanian units.

The CAGE questionnaire was used to test for alcohol abuse and dependence [23]. It consists of four short questions (‘Have you ever felt you should cut down on your drinking?’, ‘Have people annoyed you by criticizing your drinking?’, ‘Have you ever felt bad or guilty about your drinking?’, ‘Have you ever had a drink in the morning (as an eye-opener) to get rid of a hangover?’). Each question is scored with a ‘yes’ or ‘no’ response. Two or more ‘yes’ answers cause suspicion of drinking problems.

The question about smoking behaviour in the last three months had the following answers: ‘daily’, ‘on particular occasions’, ‘never’. Responses were dichotomized into ‘daily smoking’ and ‘occasional or never smoking’. Students were asked about illegal drug consumption: ‘Have you ever tried or consumed illegal drugs such as ecstasy, cocaine, LSD, cannabis products, etc.?’ The possible answers were: ‘never tried’, ‘tried once or twice’, ‘take about once a month’, ‘take about once a week’, ‘take several times a week’. Responses were dichotomized into ‘drug use’ and ‘drug non-use’ (answers: ‘never tried’ and ‘tried once or twice’).

Students were asked to answer several questions about their academic achievement. Students rated how important it is for them to have good grades at university on a four-point scale. Responses were grouped into ‘high importance’ (‘very important’ or ‘somewhat important’) and ‘low importance’ (‘not very important’ or ‘not at all important’). The question about the academic performance was: ‘How do you rate your performance in comparison with the average performance of your fellow students?’ Responses were grouped into ‘above average’ (‘much better’, better’), ‘average’, and ‘below average’ (‘worse’, ‘much worse’). Rating of academic requirements was assessed by the question: ‘How high are the academic requirements which are demanded of you at the university?’ Responses were dichotomized into ‘low’ (‘very low’, ‘quite low’) and ‘high’ (‘very high’, ‘quite high’).

Satisfaction with the studies at university was measured with the following statements: ‘I have doubts if I have chosen my study program correctly’; ‘I often feel lost in my studies’; ‘I get support from my fellow students in every respect’; ‘In general I have a good relationship with my professors’; ‘The organisation of my studies is chaotic, and it causes me problems’; ‘I don’t have a chance to introduce my ideas into my studies’; ’I enjoy my studies’. Every statement was rated on a four-point scale: ‘1—fully agree’, ‘2—partly agree’, ‘3—not completely agree’, and ‘4—disagree’. The third, fourth and seventh statements were reverse coded. The total score was calculated for every participant by summing the points of all statements. The distribution of total scores was divided into tertiles with cut-offs of 20 and 24. The first tertile ranged from 9 to 20 (‘dissatisfied’), the second—from 21 to 24 (‘moderately satisfied’) and the third—from 25 to 28 (‘very satisfied).

### 2.3. Statistical Analysis

Initial descriptive data analyses demonstrated that associations between academic achievements, satisfaction with studies and risky behaviours were similar in the studies performed in 2000, 2010 and 2017; therefore, the combined data were analysed. Some differences were observed between men and women, so descriptive data analyses were performed separately for both genders.

The categorical variables were presented as percentages and compared using the chi-square test and *z*-test with Bonferroni correction for multiple comparisons. ANOVA was used for comparison of the mean age of students between the studies. The associations of risky behaviours with academic achievements and satisfaction with studies were analysed using multivariable logistic regression analysis. A separate model was calculated for every risky behaviour adjusted for gender, age, study year, and other analysed behaviours.

Data analysis was performed using the statistical package IBM SPSS Statistics for Windows, Version 27.0 (IBM Corp.: Armonk, NY, USA, released 2020).

## 3. Results

In all studies, the proportion of female students was higher than males, 53.7% in 2000, 66.5% in 2010 and 59.1% in 2017 (*p* < 0.001). The main characteristics of the study population by study year and gender are presented in Table 1. In the 2000 study, the students were slightly younger than in 2010 and 2017 because of a change in the duration of school attendance in Lithuania. Most students declared that it is important for them to have good grades at university. The lowest proportion of male students who emphasized the importance of having good grades was in 2010, female students—in 2017. During the study period, the proportion of students who rated their academic performance above average increased, while the proportion of those who rated it below average decreased among males and females. Most students responded that academic requirements are high at the universities; however, the proportion of students who argued to the contrary increased more than three times during 17 years. Satisfaction with studies increased among male and female students.

In all study years, more female than male students answered that it is important for them to have good grades at university. Also, a higher proportion of female than male students was satisfied with their studies during the whole study period. Evaluation of academic performance differs between males and females in the 2000 and 2010 study years when more female than male students assessed their academic performance as above average.

Over the study period, the prevalence of risky behaviours among students has changed significantly (Table 1). The proportion of male and female students who used alcohol at least once a week and consumed 11 and more SAU per week declined. In males, the highest prevalence of drinking problems assessed by the CAGE questionnaire was in 2010, which decreased in 2017. Interestingly, the proportion of female students having drinking problems did not change. Daily smoking prevalence decreased significantly among male students and remained stable in female students. Drug use became more popular among males. No changes in drug use were found in females. The proportion of male students who reported a combination of drinking alcohol at least once a week and daily smoking decreased in 2017. The prevalence of those two behaviours among women increased in 2010 and declined in 2017. The proportion of students who had three risky behaviours: consuming alcohol at least once a week, smoking daily, and using drugs, was low and did not change statistically significant during the study period. 

A lower prevalence of alcohol consumption at least once a week was found in males compared to female students in the 2000 and 2010 study years, but not in the 2017 study year. The proportion of students who consumed 11 and more SAU per week and declared drinking problems was higher in males than females during the whole study period. The prevalence of daily smoking was higher in males than female students only in 2000, while a higher proportion of drug users was found in males compared to females in 2010 and 2017.

In males, risky behaviours were associated with the importance of good grades at the university (Table 2). The prevalence of all analysed risky behaviours was lower among male students who reported that it is important for them to have good grades compared to those who claimed the opposite. The highest difference was found in the prevalence of consumption of 11 and more SAU per week, drinking problems and drug use. The proportion of male students who declared drinking alcohol at least once a week and daily smoking were lower among those rating their academic performance above average compared to below average. The prevalence of alcohol consumption at least once a week was lower among male students who rated the academic requirements at the university as low compared to high and among those who were very satisfied with studies than dissatisfied.

Similar associations between risky behaviours and academic achievements were found in female students (Table 3). Reported high compared to low importance of good grades at the university was associated with a lower prevalence of drinking 11 or more SAU per week and daily smoking. The prevalence of all analysed risky behaviours, except drug use, was lower among female students rating their academic performance above average. The proportion of female students who reported drinking problems and daily smoking was lower among those very satisfied than dissatisfied with their studies.

Multivariable logistic regression analysis revealed that low importance of good grades at university was associated with higher odds of drinking alcohol at least once a week, daily smoking, drug use, and a combination of two risky behaviours (drinking alcohol at least once a week and daily smoking) (Table 4). Students who rated their academic performance below average were more likely to consume alcohol at least once a week, have drinking problems and combine frequent alcohol consumption with daily smoking than those rating their performance above average. The odds of daily smoking were higher for students who rated their performance as average compared to above average. Dissatisfaction with studies was associated with higher odds of alcohol consumption at least once a week and drinking problems.

## 4. Discussion

This study examined academic achievements, satisfaction with studies and substance use among first-year Kaunas university students in 2000, 2010 and 2017. During the study period, alcohol consumption decreased among male and female students, smoking prevalence declined in males and remained stable in females, while drug use increased among male students. Study results revealed significant associations between academic achievements and alcohol consumption, smoking as well as drug use.

Lithuania is one of the European countries with the highest alcohol consumption, which contributes significantly to health and well-being [18]. Since 2008, a number of alcohol control policies aimed at reducing alcohol consumption have been implemented in Lithuania, including bans on advertising, restricting sales hours, and increasing taxes for alcoholic beverages [24]. Those alcohol policies might be associated with the positive changes in the drinking habits of adolescents and young adults. In line with our observation, the Health Behaviour in School-aged Children (HBSC) study demonstrated the declining trends in the usage of alcoholic drinks at least once a week among 15-years old Lithuanian schoolchildren from 2010 onwards [25,26].

In general, male students consume alcohol more frequently and in higher quantities than females [10,16,27,28,29]. Our study found the differences in alcohol consumption between male and female students in 2000 and 2010 but not in 2017. During the study period, the prevalence of drinking problems identified by the CAGE questionnaire declined in males while did not change in females. Every fifth student declared a drinking problem in 2017. Similar drinking problem prevalence was reported by other authors who applied the CAGE test in student studies [30,31,32]. Data from the ESPAD survey, covering 24 European countries between 1999 and 2019, showed that temporal trends in beverage preferences among adolescents vary considerably between European countries and between boys and girls, with the high and increasing proportions of the use of cider/alcopops [33]. Thus, alcohol consumption among youth, including students, remains a major public health concern due to its effect on physical and mental disorders. In Lithuania, the minimum legal drinking age was increased from 18 to 20 years on 1 January 2018, obligating sellers to request documentation for age verification if a purchaser looks younger than 25 years old [24].

Our findings on the decrease of smoking prevalence in male students are in line with some other studies showing declining trends in cigarette smoking among adolescents and young adults [25,26,34,35]. Previous studies on the Lithuanian adult population demonstrated the increase in the smoking prevalence up to 2000, especially among women, while a decreasing trend in men and stabilisation in women has been observed afterwards [19,36]. Lithuania ratified World Health Organization Framework Convention on Tobacco Control in 2004 and introduced many marketing and availability restrictions associated with the decline in cigarette smoking prevalence [37]. At the same time, the latest evidence from Lithuania and other countries revealed that the usage of electronic cigarettes and polytobacco has been rising in young adults, leading to nicotine addiction and health risks [19,21,38,39,40].

A general trend toward an increase in cannabis/marijuana use by students is demonstrated globally [41,42]. In our study, drug use increased only in male students. The proportion of males who reported taking drugs at least once a month in 2017 was lower than in the similar study carried out in Lithuanian educational institutions in 2020 (6.6% and 15.5%, respectively) [21]. This could be explained by differences in study methodology. The latter study was performed not only in universities but also in colleges and included the students from all study years. In both studies, female students were much less likely to respond that they used drugs.

The present study found that females more often than males reported high importance of good grades, satisfaction with studies, and above-average academic performance. These findings coincide with the results received by other authors showing that female students had a higher grade point average, failed fewer subjects and more often attended all course exams than male students [28,43]. Several factors can explain the difference in academic achievements between males and females, including substance use, which has been determined more often among male than female students [9,16,17,28].

Our data revealed that frequent alcohol consumption (at least once a week) was associated with low importance of good grades at university. Furthermore, students who rated their academic performance below average and were dissatisfied with studies reported frequent alcohol consumption and drinking problems more often than those rating their performance above average and being satisfied with studies. These results are in agreement with findings from the studies conducted among students in other countries, which confirmed that hazardous alcohol consumption is linked to worse academic performance [4,16,43,44,45,46]. Students consuming alcohol at a hazardous level were late for class, missed classes, and were unable to concentrate and complete assignments in class; therefore, received lower grades, failed exams and had other academic problems. However, some studies did not find statistically significant associations between alcohol drinking and academic achievements, especially when controlling was undertaken for other factors [11,15,47]. For instance, research showed that students who consumed alcohol reported higher subjective well-being and increased self-efficacy, which might positively affect academic performance [48]. Thus, associations between alcohol consumption and academic achievements are complicated and require further investigation.

In our study, daily smoking was more common among students who reported low importance of good grades and academic performance below average, also among females who were dissatisfied with studies. Limited studies examine associations between cigarette smoking and academic achievements. Some authors reported that smoking was a negative predictor of lower grade point average in students [4,14,17]. Our data showed higher prevalence of drug use only in male students who declared low importance of good grades compared to those who argued the opposite. Low prevalence of drug use, especially among women, could explain week association between this risky behaviour and academic achievements. Most studies, analysing marijuana use and academic performance, found negative association [17,49,50,51]. Marijuana use adversely affected college academic outcomes through poorer class attendance [49]. Frequent marijuana users were more likely to drop out of college and plan to delay graduation when compared to non-users, also they reported lower grade point average than non-users [50]. A negative association between momentary craving for marijuana and academic motivation was found [17].

Evidence suggests that students tend to combine multiple unhealthy behaviours: use of alcohol, tobacco and cannabis, which might have a much greater impact on academic achievement and health than a single-behaviour [11,14]. Our study found that the proportion of students who combined frequent alcohol consumption with daily smoking decreased in 2017. A combination of two risky behaviours was associated with low importance of good grades at university and rating their academic performance below average. A study carried out in Spain showed a significant negative effect of the combined binge drinking and cannabis consumption, but not of binge drinking alone, on grade point average and academic adjustment [11]. The combined consumption of alcohol and cannabis led to difficulties in adaptation to academic life, which contributed to poorer performance at university. A study analyzing longitudinal alcohol and marijuana use among college students found that students who used both substances moderately or heavily received lower grades than students who consumed little or no substances [14]. The prevalence of a combination of three risky behaviours: consuming alcohol at least once a week, smoking daily, and using drugs, was low in our study because of the low proportion of drug users.

Strengths and limitations. Our study has some strengths and limitations. The study was performed in large representative samples of first-year students from all Kaunas universities and different study programmes. It covered 17 years and used the same methodology, which allowed us to demonstrate how academic achievements and risky behaviours have changed over time. There is a lack of such data on Lithuanian students. Furthermore, we analysed the wide range of risky behaviours such as alcohol consumption, smoking and drug use and their associations with academic achievements among first-year university students. The majority of previous research analysed the association of isolated unhealthy behaviour and academic performance among students only at a one-time point.

Several limitations also should be mentioned. The study was carried out only in the universities of Kaunas, which is the second largest city in Lithuania with a large student community; however, it cannot represent all Lithuanian students. The data was collected by using self-administered questionnaires, which can lead to an under-or over-estimation of risky behaviours. Because information about risky behaviours can be sensitive personal matters or stigmatized, students’ responses may be subject to social desirability bias. Anonymity of the questionnaire can help reduce this problem. For academic achievement analysis, we used self-reporting information about the importance of good grades, academic performance and requirements and satisfaction with studies without objective measurements such as grade point average or the number of failed subjects. Finally, our study was cross-sectional, so we cannot detect causal relationships.

## 5. Conclusions

This is the first study conducted in Lithuania showing trends in academic achievements and risky behaviours among first-year university students and their associations. Despite the decline in alcohol consumption and smoking, these risky behaviours remain prevalent among students, and the prevalence of drug use is increasing. Risky behaviours were associated with worse academic achievements and dissatisfaction with studies. Health promoting interventions at the individual and student community level are required to reduce the prevalence of substance use and improve the academic achievements of students. Further studies exploring factors associated with risky behaviours and their effect on academic performance are needed to select more targeted preventive measures.

## Figures and Tables

**Table 1 ijerph-19-07616-t001:** Characteristic of the study population in 2000, 2010 and 2017 study years.

Characteristic	Males				Females			
	2000	2010	2017	*p*-Value between Years	2000	2010	2017	*p*-Value
	(*n* = 477)	(*n* = 338)	(*n* = 526)		(*n* = 554)	(*n* = 670)	(*n* = 760)	between Years
Age (years, mean, SD)	19.7 (1.3)	20.1 (1.1)	20.2 (1.2)	<0.001	19.7 (1.7)	20.1 (1.1)	20.2 (1.1)	<0.001
Importance of good grades (%)								
High	93.5 #	83.8 *#	89.5 #	<0.001	96.7 *	96.1 *	93	0.003
Low	6.5	16.2	10.5		3.3	3.9	7	
Academic performance (%)								
Above average	25.2 *#	30.8 *#	40	<0.001	34.4 *	35.5 *	43.3	<0.001
Average	57.4 *	54.8	49.5		53.3	55.7	49.7	
Below average	17.4 *	14.4 #	10.5 #		12.3 *	8.7	7	
Academic requirements (%)								
Low	8.0 *	10.5 *#	24.6	<0.001	6.3 *	5.9 *	21.2	<0.001
High	92	89.5	75.4		93.7	94.1	78.8	
Satisfaction with studies (%)								
Very satisfied	18.8 *#	28.1 #	28.3 #	<0.001	25.7 *	38.8 **	33.7	<0.001
Moderate satisfied	33.5	36.9	39.3		35.9	36	34.9	
Dissatisfied	47.7 *#	35.0 #	32.4		38.5 *	25.2 *	31.4	
Drinking of alcohol at least once a week (%)	61.7 *#	53.6 *#	29.5	<0.001	38.3 *	40.3 *	26.2	<0.001
Drinking of ≥11 alcoholic units a week (%)	13.3 #	16.2 *#	9.7 #	0.024	6.3 *	5.5 *	1.6	<0.001
Drinking problems (CAGE scores ≥ 2) (%)	25.5 #	37.3 *#	27.4 #	0.001	17.4	21	21.1	0.198
Daily smoking (%)	29.5 *#	21.1 **	18.5	<0.001	18.8	19.2	16.1	0.251
Drug use (%)	0.7 *	5.5 #	6.6 #	<0.001	0.8	0.8	0.8	0.997
Drinking of alcohol at least once a week and daily smoking (%)	20.4 *#	18.3 #	9.9	<0.001	8.8	13.7 *	8.7	0.004
Drinking of alcohol at least once a week, daily smoking and drug use (%)	0.2	1.6 #	1.2	0.166	0.4	0.3	0.4	0.949

** p* < 0.05 compared with 2017 study year (*z* test with Bonferroni correction); ** *p* < 0.05 compared with 2000 study year (*z* test with Bonferroni correction), # *p*< 0.05 compared with female students in corresponding study year.

**Table 2 ijerph-19-07616-t002:** The proportion (%) of risky behaviours according to academic achievements and satisfaction with studies in male students.

Academic Achievements and Satisfaction with Studies	Drinking Alcohol at Least Once a Week		Drinking of ≥11 Alcoholic Units a Week		Drinking Problems (CAGE Scores ≥ 2)		Daily Smoking		Drug Use	
	%	*p*-Value	%	*p*-Value	%	*p*-Value	%	*p*-Value	%	*p*-Value
Importance of good grades		0.01		<0.001		0.001		0.049		<0.001
High	45.4		10.9		27.7		22.3		3.5	
Low	57.4		26.6		41.2		29.7		10.4	
Academic performance		<0.001		0.112		0.126		0.024		0.446
Above average	36.3 *		9.8		26.1		18.8 *		5.2	
Average	49.4		13.9		29.9		24.7		3.7	
Below average	61		14		34.1		27.3		4	
Academic requirements		0.013		0.381		0.737		0.192		0.071
Low	38.5		10.6		28.2		19.5		6.6	
High	48.2		12.9		29.4		23.7		3.8	
Satisfaction with studies		0.003		0.958		0.442		0.914		0.446
Very satisfied	41.6 *		12.8		26.7		23.6		3.5	
Moderate	43.1		12.6		29.8		22.4		3.6	
Dissatisfied	52.4		12.1		30.9		22.5		5.2	

* *p* < 0.05 compared with the worse academic performance or dissatisfaction with studies (*z* test with Bonferroni correction).

**Table 3 ijerph-19-07616-t003:** The proportion (%) of risky behaviours according to academic achievements and satisfaction with studies in female students.

Academic Achievements and Satisfaction with Studies	Drinking Alcohol at Least Once a Week		Drinking of ≥11 Alcoholic Units a Week		Drinking Problems (CAGE Scores ≥ 2)		Daily Smoking		Drug Use	
	%	*p*-Value	%	*p*-Value	%	*p*-Value	%	*p*-Value	%	*p*-Value
Importance of good grades		0.199		0.031		0.13		<0.001		0.136
High	33.9		4		19.7		17.2		0.7	
Low	40.2		8.6		26		31.6		2.1	
Academic performance		<0.001		0.001		0.046		0.002		0.182
Above average	29.8 *		3.4 *		19.4		14.6 *		0.4	
Average	35.5		3.9		19.2 *		19		0.9	
Below average	45.1		9.6		27.1		25		1.7	
Academic requirements		0.419		0.569		0.084		0.906		0.845
Low	31.8		3.5		24.2		17.6		0.9	
High	34.4		4.3		19.4		17.9		0.8	
Satisfaction with studies		0.11		0.086		0.033		0.03		0.904
Very satisfied	30.8		3.3		16.6 *		14.7 *		0.6	
Moderate	35.8		3.9		21.3		18.9		0.7	
Dissatisfied	35.4		5.7		22		20.2		0.9	

* *p* < 0.05 compared with the worse academic performance or dissatisfaction with studies (*z* test with Bonferroni correction).

**Table 4 ijerph-19-07616-t004:** Odds ratios * (95% CI) of risky behaviours by academic achievements and satisfaction with studies.

Academic Achievements and Satisfaction with Studies	Drinking Alcohol at Least Once a Week		Drinking Problems (CAGE Scores ≥ 2)		Daily Smoking		Drug Use		Drinking Alcohol at Least Once a Week and Daily Smoking	
	OR	*p*-Value	OR	*p*-Value	OR	*p*-Value	OR	*p*-Value	OR	*p*-Value
	(95% CI)		(95% CI)		(95% CI)		(95% CI)		(95% CI)	
Importance of good grades		0.041		0.1		0.010		0.016		0.005
High	1		1		1		1		1	
Low	1.38		1.3		1.54		2.22		1.66	
	(1.01–1.88)		(0.95–1.79)		(1.11–2.12)		(1.16–4.24)		(1.16–2.36)	
Academic performance										
Above average	1		1		1		1		1	
Average	1.31	0.002	1.05	0.589	1.24	0.04	0.86	0.593	1.51	0.001
	(1.10–1.56)		(0.87–1.27)		(1.01–1.35)		(0.49–1.51)		(1.18–1.93)	
Below average	1.8	<0.001	1.45	0.009	1.3	0.102	0.85	0.709	2.18	<0.001
	(1.37–2.35)		(1.10–1.92)		(0.95–1.77)		(0.36–2.00)		(1.56–3.08)	
Satisfaction with studies										
Very satisfied	1		1		1		1		1	
Moderate	1.07	0.514	1.32	0.016	1.11	0.384	0.7	0.278	1.11	0.447
	(0.88–1.31)		(1.05–1.65)		(0.88–1.41)		(0.36–1.34)		(0.85–1.46)	
Dissatisfied	1.23	0.048	1.38	0.006	1.11	0.399	0.76	0.414	1.13	0.347
	(1.01–1.51)		(1.10–1.74)		(0.87–1.41)		(0.39–1.48)		(0.86–1.49)	

* Separate logistic regression analysis was performed for every risky behaviour. Odds ratios were adjusted for gender, age, study year, and other analysed risky behaviours; OR-odds ratio; CI-confidence intervals.

## Data Availability

The data presented in this study are available on request from the corresponding author. The data are not publicly available due to ethical issues.

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
