# Peer review of "Academic Achievements, Satisfaction with Studies and Risky Behaviours among First-Year Students of Kaunas (Lithuania) Universities, 2000–2017"

_ijerph, 2022, doi:10.3390/ijerph19137616_

Round 1
Reviewer 1 Report
The article is coherent, although it needs a review of its writing. The topic is very interesting and covers a relevant topic. At first glance, it seems to offer a significant perspective, with a great contribution to the educational community. However, it has a research design that can be considerably improved to meet the scientific rigor required by the journal.
It is recommended:
Check spelling and grammar.
Respect IMRyD structure in the abstract. Data discussion is missing.
Restructure the introduction. The theoretical foundations studied should be deepened. It is recommended to rewrite the introduction and write new sections in which what has been investigated to date is made clear. These findings should serve to justify the conclusions of the study and discussion of results.
Review MDPI citation and referencing in depth.
Justify the use of statistics and clarify in the abstract.
Provide instrument validation data using a Confirmatory Factor Analysis (CFA). The Unweighted Least Squares technique is recommended.
Try to link the initial reflections (theoretical review) with the discussion and conclusions.
Add study limitations. In the same way, possible future lines of research should be added.
Reviewer 2 Report
The study focused on the analysis of academic achievements satisfaction with studies and risky behaviors among students. The article is interesting and covers very important aspects of students’ nutritional behavior. Risk-taking behaviors have negative consequences on young adult’s health, therefore addressing this issue seems highly justified. It is also important to note that the study includes a 17-year follow-up period, which provides an opportunity to compare students' risky behaviors over time.
The structure of the article is logical and the sections are written appropriately.
I have only a few minor suggestions:
- In the results section, some results are unnecessarily presented twice (in tables and in the text). Please consider shortening the text on lines 188-200 and presenting only the most important data.
- Consider using term drinking problem instead of problem drinking
- As reported in lines 341 students tend to combine multiple unhealthy behaviors, therefore it would be useful to report the number of students stating all behaviors defined as risky during the analyzed time period.
- The text between lines 368-371 should be deleted.
- The authors must keep in mind that the data obtained are student declarations, thus drawing conclusions must be done with caution.
In conclusion, the manuscript should be qualified for publication pending a minor revision
Reviewer 3 Report
The manuscript consists of total 12 pages, including 4 tables and the list of total 48 literature references. The text has the structure typical for scientific publications, although it would be advisable to make the strengths and weaknesses description into a separate section instead of keeping it in the end of the Discussion section. The article presents original results of the questionnaire-based study on the prevalence of risky behaviors among the studying youths. As such, it is both current and interesting problem, fitting into the scope of works published in the Journal. The article presents as a pleasurable read as it has a logical line of argumentation and is written in correct communicable English. The title of the manuscript is adequate to its contents.
The Abstract mirrors the key contents of the main text properly.
The Introduction section presents the background of the raised topic adequately.
The Material and methods section is clear and detailed enough.
The Results section presents study outcomes that are consistent with the methodology by the Authors, clearly and convincingly; it is adequately aided with tables.
The Discussion places the presented results into the context of earlier published knowledge in the field adequately.
The Conclusion is based on the discussed results.
The References are numerous and recent enough, relevant to the topic of the manuscript.
However, the Authors may consider adding some mentions to their article on the aspects of chosen other countries youth lifestyle, e.c. in https://doi.org/10.3390/ijerph182010746
https://doi.org/10.3390/ijerph18052548
https://doi.org/10.3390/socsci9110187
https://doi.org/10.3390/ijerph16060965
https://doi.org/10.3390/ijerph19063591
https://doi.org/10.3390/ijerph182010933
